# Smart Quantum Tunneling Composite Sensors to Monitor FKM and FFKM Seals

**DOI:** 10.3390/s23031342

**Published:** 2023-01-25

**Authors:** Mookkan Periyasamy, Carmen J. Quartapella, Nicholas P. Piacente, Gary Reichl, Brian Lynn

**Affiliations:** GT Services, LLC, 1684 South Broad Street, P.O. Box 1307, Lansdale, PA 19440, USA

**Keywords:** FKM and FFKM compounds, carbon nanostructure, intrinsic sensing QTC-based seals, O-rings and components, monitoring, lifetime prediction, performance tracking

## Abstract

Operators of industrial machinery relentlessly pursue improving safety, increasing productivity, and minimizing unplanned downtime. Elastomer seals are ubiquitous components of this machinery. In general, static seals are designed to be compressed at a fixed level of compression, taking gland geometry, loading condition, temperature range of operation, fluid media exposure, and other factors into account to ensure the safe operation of equipment. Over time, seals experience compression set, chemical-induced swelling, erosion, and other phenomena which can compromise the compressive force generated by the seal and cause leaking. This is particularly important in critical applications, where high pressure, high temperature, and aggressive media are present, and fluorinated elastomers are common materials for seals. Further, changes in operating conditions at manufacturing plants, either intentional or through regular process variation, create unknown operating conditions for seals. This unknown and variable application environment makes seal performance hard to predict. Therefore, machinery utilizing seals is, at best, serviced preventatively at certain intervals, where seals are removed, and the remaining useful life of the seal is unknown. This leads to unnecessary machinery downtime and increases consumable costs for manufacturers. In the worst case, the seal is run to failure, creating machinery and plant safety concerns. Both scenarios are undesirable for manufacturers using industrial machinery. This paper reports on the development of “smart” intrinsic self-sensing seals, which enable performance monitoring of the compression behavior of seals while in use. In addition, this paper examines quantum tunneling elastomeric composites (QTC) to demonstrate a method of component performance monitoring by modifying the underlying elastomeric material itself. This paper studies QTC sensor-based fluorinated (FKM) and per-fluorinated (FFKM) compositions, which are modified to incorporate varying levels of carbon nanostructure (CNS) material. The resulting seal’s resistive properties are shown to be a function of the level of compression, the first time this phenomenon has been demonstrated in high-performing FKM and FFKM seal materials.

## 1. Introduction

A composite is produced from two or more constituent materials to achieve better performance than individual components [1]. They are classified into four categories based on the matrix constituent, such as (a) polymer matrix composites, (b) metal matrix composites, (c) ceramic matrix composites, and (d) quantum tunneling composites [2,3,4,5,6,7,8,9,10,11,12,13,14,15,16,17,18,19,20,21,22,23,24,25,26,27,28,29,30]. Quantum tunneling composites (QTCs) are combinations of polymer composites having both dielectric elastomer and conductive metal particles. In the QTC’s inactive state, the conductive elements are too far from one another to pass electron charges. When sufficient force is applied, the electrons start to flow without the particles touching each other due to the spiky nature of the conductive material, and the amount of electric current passed exponentially proportional to the applied force on QTC. The electric current stops when the force is taken away in QTC. Within the finished structure of QTC, the individual elements remain separate and unique, distinguishing composites from mixtures and solid solutions. QTCs are demonstrated to use different applications, such as smart textiles, foot wears, touch pads, display devices, NASA’s Robonaut, mp3 players, sporting materials, automotive, and life science applications, such as measuring blood pressure, making safe batteries, and extreme weather conditions [5,6,7,8,9,10,11,12,13,14,15,16,17,18,19,20,21,22,23,24,25,26,27,28,29,30]. The QTC pills for such applications were generally made using primarily silicone as a non-conductive elastomer and Nickel as one of the conductive materials. Additional details for the compositions of QTCs and their uses can be obtained from the literature reports and patents. However, there are no literature or patent reports on using fluoro- (FKM), or perfluoro- (FFKM) compounds as elastomers and carbon nanostructure (CNS) as the conductive material to make self-sensing QTC-based seals or gaskets and using them to monitor their own lifetime for the products. FKM and FFKM-based seals and gaskets are relatively very expensive.

Fluoroelastomer (FKM) and perfluoro-elastomer (FFKM) are commonly used for seals, gaskets, and other components for various applications, such as Aerospace and Defense, Oil and Gas, Semiconductor, Life Sciences, Chemical Processing, Agriculture, and Industrial Operations. The FKM and FFKM-based seals were manufactured by formulating FKM and FFKM-based compounds having curatives, fillers, and other additives. For certain applications, the fillers include conductive metal or carbon powder and/or particulate fillers and/or fiber fillers, which may include metals, polymers, carbon, graphite, and similar materials.

The generic compositions and requirements of FKM and FFKM compounds used for manufacturing various seals, gaskets, and components, particularly those for use in semiconductor manufacturing, including dynamic valve seals, static O-rings, and other seals and gaskets, are well known [31,32,33]. Semiconductor manufacturing equipment is kept vacuum-tight with seals or gaskets to ensure that contaminants remain outside the reaction chamber and that reactants from within the chamber do not escape the chamber.

A goal for seals is to maximize their useful life in the harsh conditions within the reaction chamber, as these reaction vessels process high-temperature, aggressive plasma to attack elastomers. In some cases, as seal material degrades and erodes, sealing integrity can be compromised, and the reaction chambers will not be able to sustain the desired vacuum levels needed for the process. This can lead to more frequent maintenance, more equipment downtime, and reduced production output. Thus, maximizing useful seal life is a highly desired goal in most semiconductor manufacturing applications.

Measuring the reliability and degradation rates of seals and gaskets also is a desirable end objective in a wide variety of end applications to understand the effective production life of a seal or a gasket while minimizing the leaks or end effects of compromised seal properties due to degradation. Particularly in very aggressive applications, maximizing useful seal life will only prolong the inevitable replacement of such worn components. In this case, understanding the ideal time to replace such a component is required to realize any benefits of longer-lasting seals.

Such seals are generally formed of highly chemically-resistant elastomeric materials that are costly to purchase, so it is understandable that one may not want to replace a seal before replacement is indicated. At the same time, unplanned maintenance downtime due to equipment failure can be even more costly to a manufacturer than the replacement of the relatively inexpensive consumable [34]. Therefore, a point exists where optimum cost savings can be achieved by scheduling equipment maintenance at the ideal point. Broadly, predictive maintenance models have been developed in an attempt to find this point [34,35,36] with the ultimate goal of enhancing yield, reducing unplanned downtime, and general improvement of manufacturing efficiency.

One common approach to implementing predictive maintenance is to ‘sensorize’ equipment in an attempt to connect specific sensor output to critical performance metric information. For example, the use of vibration sensors to measure bearing performance inside a motor, where an optimal sensor location is found, and time-series vibration signals are analyzed to determine the point-of-failure of the bearings in industrial applications [34,36]. One other such example for maintenance of seals is a slit-valve door (BSV) seal described in a U.S. Patent [37], which is based on the use of sensors positioned on various locations of a slit valve door seal (Bonded Slit Valve, also known as BSV seal) that used external strain sensor data attached to the sealing product (along with other factors) to understand product lifetime, thus creating a seal lifetime monitoring system. Such attempts to measure properties are highly useful; however, some applications of predictive maintenance for consumables may not be sensitive enough (or at all) to any external sensor, meaning that a more thorough correlation between failure mode and sensing mechanism needs to be developed. In addition, some extrinsic sensing schemes rely on a detection mechanism that is an output of failure (i.e., vibration resulting from a failed bearing) and not a predetermining factor of failure. In addition, seal life that does not require significant sensor usage to provide needed data would be advantageous in the art, particularly if such sensors did not have a significant impact on the ability of the seal to function and maintain consistent properties over the seal without the use of embedded microsensors or other sensor devices attached to the seal.

There remains a need in the art for improved ways to ensure maximum useful seal life and critical analysis that would assist in choosing the optimal seal and associated maximum seal life for use in manufacturing facilities so as to minimize downtime, improve maintenance cycles and avoid seal degradation failures, which is simple to monitor and which, preferably, does not impact the operation of the seal or the consistency of its properties, including maintaining good seal elastomeric, mechanical and chemically resistant properties.

This paper describes a method of monitoring seal life where the sensing mechanism is intrinsic to the elastomeric material. Specifically, the use of a QTC technology where levels of conductivity are consistent through the seal and can range from low to high levels of conductivity and are a function of seal compression. Such materials would be particularly useful in semiconductor applications where low levels of particulation and contamination are desired, and electrical output can be monitored and tied to the level of compression of the seal. CNS additives have a “spikey needle-type structure” and are chemically similar to carbon black additives used in traditional formulations, so it is expected that chemical properties of materials, such as erosion, corrosion, etc., will not be significantly affected.

## 2. Experimental: Materials and Methods

### 2.1. Raw Materials

Fluoro-elastomers (FKM) and Perfluoro-elastomers (FFKM) can be purchased from supplying companies such as Chemours, 1001 N. Market St, Wilmington, DE 19801, USA, Daikin, 20 Olympic Dr. Orangeburg, NY 10962, USA, DuPont, 4417 Lancaster Pike, Wilmington, DE 19805, USA, and Solvay, 10 Leonard Ln, West Deptford, NJ 08086, USA. Carbon N-990 Black is available as Thermax^®^ and was purchased from Cancrab, 1702 Brier Park, Crescent N.W, Medicine Hat, Alberta, Canada T1C 1T9. Austin Black 325 can be purchased either from Harwick Standard Company, 60 S. Seiberling St, Akron, OH 44305, USA, or CFI Carbon Products Company, P.O. Box 1065, Bluefield, VA, USA. CNS material can be purchased from Cabot Corporation, 4400 North Point Parkway, Suite 200, Alpharetta, Georgia 30022, USA, or one of its distributors. DAIC DLC was purchased from Natrochem, Inc. P.O. Box 1205, Savannah, GA 31402-T205. Varox DBPX-50 was purchased from R.T. Vanderbilt Company, Inc., 30 Winfield St, P.O. Box 5150, Norwalk, CT 06856-5150, USA. Finally, 2,2-bis[3-amino-4-hydroxyphenyl] hexafluoropropane, also known as diaminobisphenol AF or BOAP, can be purchased from Sigma Aldrich Chemical Company, 3050 Spruce St, St. Louis, MO 63103, USA or TCI Americas, 9211 N. Harborgate St, Portland, OR 97203, USA.

### 2.2. Sample Preparation

The QTC elastomer composite compositions were formed using Solvay FKM Tecnoflon^®^ VPL X75545 Fluoro-elastomer or PFK-300 FFKM Perfluoro-elastomer from Lodestar, 8 Arbor Dr, Howell Township, NJ 07731, USA, which is curable and was cured using BOAP as curative. The CNS in pellet form was incorporated in varying levels by weight based on 100 parts by weight of the Tecnoflon^®^ base polymer. CNS, other fillers, and curative materials (respective of each formulation) were stirred until all ingredients were thoroughly mixed. Then, these ingredients were combined with the polymer and mixed in a mill for up to 30 min to create a homogenous compound in sheet form, using traditional mill and mixing techniques. Compound sheets were prepared from the milled material using click-and-die procedures in which the material is cut with an ID/OD clicker. The die-cut material was then cut to length and wrapped in a compression mold. FKM formulations were made into −214 size O-rings and button samples. FFKM formulations were made into −326 size O-rings.

### 2.3. Test Methods

Volume resistivity is calculated from geometric factors and experimentally-derived resistivity measurements. For resistance experimentation, a four-point probe method, well-known in the literature [38,39], is used. The four-point probe technique involves bringing four probes in contact with a material of unknown resistance. A direct current is forced between the outer two probes, and a voltmeter measures the voltage difference between the inner two probes. A Keithley 2410 Source meter was used to carry out the four-point probe measurements. To eliminate sources of variation (i.e., contact resistance between components, irregularity between compression, etc.), a fixture, as shown in Figure 1, was used to compress the seal uniformly and measure the conductivity of the seals at various levels of seal compression. The fixture uses copper plates insulated with Teflon^®^ to connect the samples to the Source meter and a vice to compress the seals to various levels of compression. The four-point probe measurement was carried out in constant current mode, set to 100 mA. The samples were placed in the vice and closed to have light contact with the fixture for the 0% compression levels.

It’s a general rule of thumb to have seals compressed at a minimum of 12–25% compression depending on the temperature of application, dynamic conditions, seal mechanical properties, and other factors. To mimic application conditions, the compression was increased in 5% increments, and the conductivity was measured to a maximum of 30% compression, which is a reasonable maximum level of compression known in the elastomer sealing industry [40].

## 3. Results and Discussion

### 3.1. Formulations

Compositions of samples 1–9 are shown in Table 1. FKM (Fluoroelastomer) is used in seal formulations to have chemical and corrosion resistance for reliable performance under harsh environmental conditions. N-990 black and Austin carbon black are included as fillers in the seal formulations to achieve ideal mechanical properties. The ingredients TAIC DLC and Varox DPBH-50 are crosslinkers and are present in the seal formulation to vulcanize the seal. Carbon Nanostructure (CNS) acts as an added filler in addition to offering higher conductivity than other known fillers. Testing of samples having 0–4.8 parts per hundred (PHR) by weight of the CNS additive per 100 parts by weight of curable polymer demonstrated in initial screening that a level of at least about 4.8 PHR of the CNS additive in the formulations provided an amount adequate for measuring volume resistivity. Further testing was carried out using samples 12–20 in the form of O-rings and buttons with carbon nanostructure additive present in amounts of from 4.8 to 9.6 PHR of the curable polymer.

### 3.2. Volume Resistivity of O-Rings

The first round of samples 1 to 5 was tested to baseline the effects of the addition of CNS additive to the O-Rings, using between 4 and 9.6 PHR. In general, the volume resistivity was found to decrease as the content of the CNS additive increased. This is expected because increased CNS additives should shorten the conductive pathway through the sample, reducing resistance. Sample 2 has lower volume resistance than Sample 3, which is possibly an outlier or due to variation in sample preparation or testing. The QTC effect was apparent once a load was applied. When 5% deflection was applied, as also shown in Figure 2, volume resistivity decreased by 4.5 times for Sample 1, having 4.8 parts by weight of the additive. Similar trends were observed for Samples 3 to 5. In most of the formulations, volume resistivity starts to become constant, after about 5–7% compression, possibly due to a minimum conductive pathway being reached through the material after the 5–7% compression mark, which results in a lack of sensitivity to compression after this point.

The baseline testing revealed an order of magnitude difference between the first sample and the other four samples at 0% compression. Sample 5 (9.6 PHR CNS) shows a volume resistivity of ~33 Ohm-cm, which is due to the relatively high loading of conductive CNS content within the sample. Sample 5 has the lowest resistivity (33 Ohm-cm) versus Sample 1 (1157 Ohm-cm). With a goal of maximizing sensing ability, it would be ideal to have a large change in volume resistivity as a function of seal compression. Therefore, to maximize the change in volume resistivity across the usable compression range of the seal, it was determined that an optimal range of CNS filler likely exists no higher than 6 PHR and, possibly, below 4.8 PHR. Therefore, a second round of Samples (6–9) was run through testing to characterize the range between 4 and 4.8 PHR. These results are shown in Figure 3 for Samples 6 to 9. Sample 9 is the same formulation as Sample 1 but was prepared and tested independently with Samples 6 to 8.

The second round of samples confirmed the trend of increasing CNS content with reduced resistivity at 0% compression. Although the two repeat trials at 4.8 PHR CNS content (mixed from different batches) show nominally different values, this could be due to a combination of experimental setup variation and variability in the batches, considering the relationship between conductivity and CNS content remains intact. The results indicate that as low as four PHR CNS in the FKM formulation can produce QTC-based self-sensing properties suited for O-ring monitoring applications.

### 3.3. Volume Resistivity of Buttons

Volume resistivity of QTC-based buttons: When testing the buttons, and with reference to Figure 4, button Samples 10 to 14 were tested, and made from the compositions of Samples 1 to 5 one can notice that the volume resistivity decreases when the CNS additive is doubled from 4.8 PHR to 9.6 PHR.

At a constant level of CNS additive, the volume resistivity initially decreases as the deflection is increased from 5 to 10%. When the deflection is increased beyond this range, the volume resistivity increases. The state of strain of the CNS particles within the button is quite non-linear in this range of compression. Considering the four-point measurement is merely a measure of the most conductive pathway through the material at a given time, one possible explanation for this behavior is that applying compression beyond the 5–10% range is most likely pushing the conductive CNS particles away from each other, which, in turn, reduces the conductivity. The result also indicates that the geometry of seals plays a role in the QTC effect, considering the relationship for O-rings (Figure 2) indicates that volume resistivity remains constant at higher levels of compression. The authors plan to do further research to understand the comparative performance of O-Rings versus button-type seals in the future.

### 3.4. Volume Resistivity of FFKM O-Rings

Volume resistivity of QTC-based KF Seals: The KF seals were prepared using compound formulations having carbon nanostructure additive in accordance with the ISO Standard 2861-1 size ISO40 (referred to herein as “KF-40” seals). These seals were subjected to conductivity testing. QTC FFKM formulations were prepared using the procedures (i.e, in Experimental 2.2 and tested as in Experimental 2.3 for FKM O-Rings) to identify a preferred composite formulation to prepare a KF-40 composite seal. The Cabot CNS in pellet form was incorporated in varying levels in ranging from as measured in parts per hundred (PHR) by weight per 100 parts by weight of the PFK-300 base polymer. Additional fillers used in the first five samples (Samples 15, 16, 17, 18, and 19) included 3 PHR of the Austin Black 325, and 5 PHR of the conductive thermal carbon black N-990 from Cancarb, 1702 Brier Park, Crescent N.W, Medicine Hat, Alberta, Canada T1C 1T9, available as Thermax^®^. The remaining Samples (Samples 20–23) were formed in the same manner as Samples 15–19 but did not include the Austin Black 325 or the Thermax^®^ N-990 additives. Instead, the only filler was the Cabot carbon nanostructure pellets in amounts ranging from 8 to 14 PHR of the base polymer PFK-300.

Testing was initially performed on samples 15–19 per the procedures outlined earlier in the experimental section. Samples 15 and 16 (4 and 6 PHR CNS, respectively) did not register any conductivity at 0% compression and were not tested any further. The results from the rest of the samples (17–19) can be seen in Figure 5 below.

As the various samples are compressed, the resistivity decreases until approximately 10% compression and then starts to increase. This is in line with the results achieved from experimentation with the FKM buttons, where increasing the content of CNS within the formulation reduces the overall volume resistivity. In addition, the result highlights the impact of sample geometry on the compression versus resistivity relationship. The overall findings indicate that greater than 6 PHR of CNS material is needed in the formulation to obtain a conductive sample at 0% compression with the chosen base FFKM polymer.

### 3.5. Volume Resistivity of FFKM O-Rings without Carbon Black Fillers

To isolate the impact of the CNS material on the resistivity versus compression effects, several samples (samples 20–23) were formulated without the Carbon Black fillers. In other words, only CNS material was used as filler content for the QTC elastomer samples. Figure 6 demonstrates samples having only CNS additives in amounts from 8, 10, 12, and 14 PHR. Each formulation test included four repeats to verify experimental repeatability.

A larger resistivity change across the compressive region is shown in samples that are more highly loaded with the CNS content, indicating that the relationship between resistivity and sample compression remains intact, with or without the Carbon Black fillers present in the formulation. The 14 PHR CNS formulation (sample 23) shows the highest change in resistivity up to approximately 15% compression. Above 15% compression, there is a limited sensitivity to compression change. This sensitivity profile is improved over others, considering 15% compression is adequate for many O-ring sealing applications, and lack of sealing force can cause O-ring failure as the seal approaches 0% compression. This indicates that sample 23 is the most sensitive and most viable formulation tested for QTC sensing applications.

The replacement of CNS filler for other conductive fillers can be seen by comparing this result to the other filled sample at 14 PHR (sample 16), for an apples-to-apples comparison of conductive filler quantities. Sample 16 (8 PHR CNS + 6 PHR carbon blacks) did not register a conductivity at 0% compression, compared to sample 23, which was the most sensitive formulation tested. Therefore, a clear benefit can be seen for using the CNS material only for resistive change QTC applications.

## 4. Conclusions

QTC-based “smart” seals were made from FKM and FFKM-based compounds, formulated with CNS material and other additives. Four-point probe test method was used to measure the volume resistivity of QTC-based samples, including various seal and button geometries. The use of CNS material in QTC seal formulations is shown to increase sensitivity of elastomer electrical properties to seal compression, relative to lesser-filled formulations or similarly filled formulations with other conductive filler schemes. The material formulations are useful for critical applications, where performance-monitoring of seals is most important, and FKM and FFKM-based seals are used. This can help to eliminate unnecessary machinery downtime and reduce consumable costs for manufacturers, plus to operate the machinery safely.

The relationships outlined in this paper indicate a large dependence on sample geometry. More research needs to be done to understand the impacts of sample geometry on the QTC effects.

## Figures and Tables

**Figure 1 sensors-23-01342-f001:**
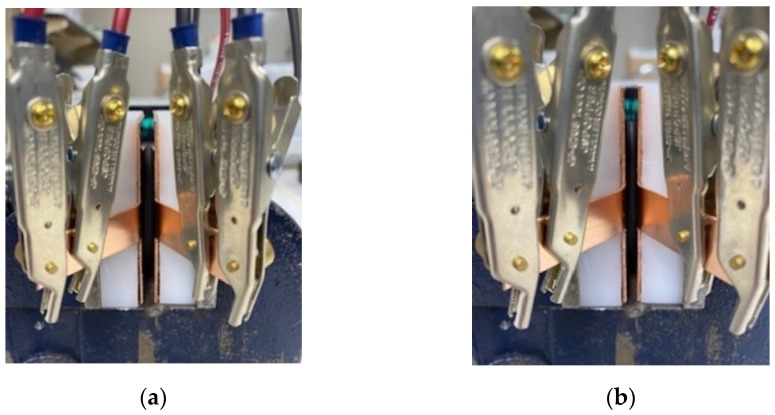
(**a**) Experimental setup for O-ring resistivity testing at 0% compression. (**b**) Experimental setup for O-ring resistivity testing at 30% compression.

**Figure 2 sensors-23-01342-f002:**
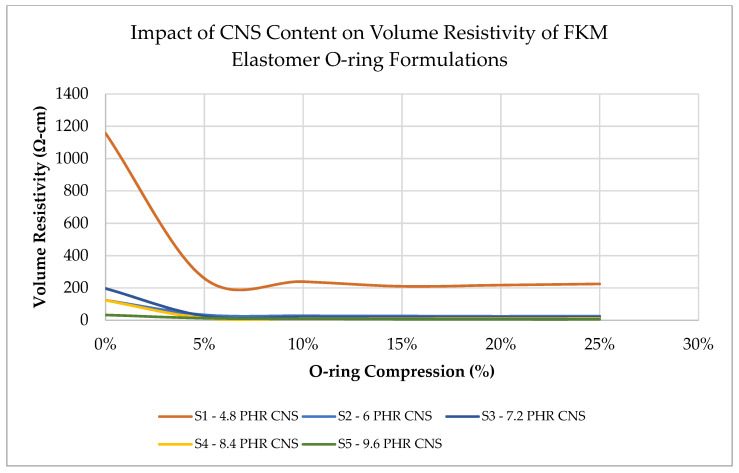
Effect of CNS content in the range of 4.8 to 9.6 PHR within elastomer O-rings on volume resistivity across multiple compression levels.

**Figure 3 sensors-23-01342-f003:**
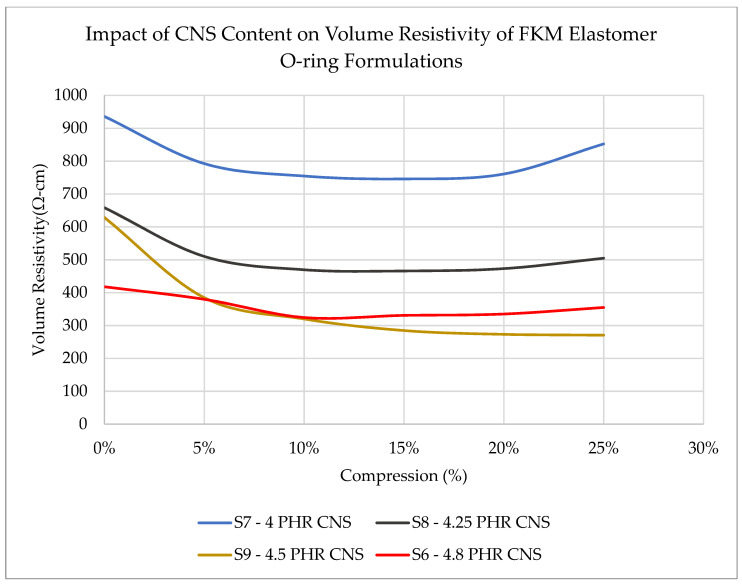
Effect of CNS content in the range of 4.0 to 4.8 PHR within elastomer O-rings on volume resistivity across multiple compression levels.

**Figure 4 sensors-23-01342-f004:**
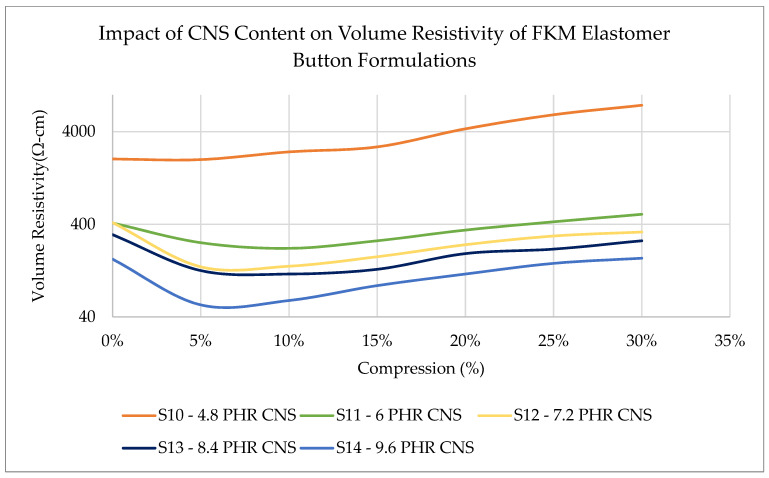
Effect of CNS content within elastomer buttons on volume resistivity across multiple compression levels.

**Figure 5 sensors-23-01342-f005:**
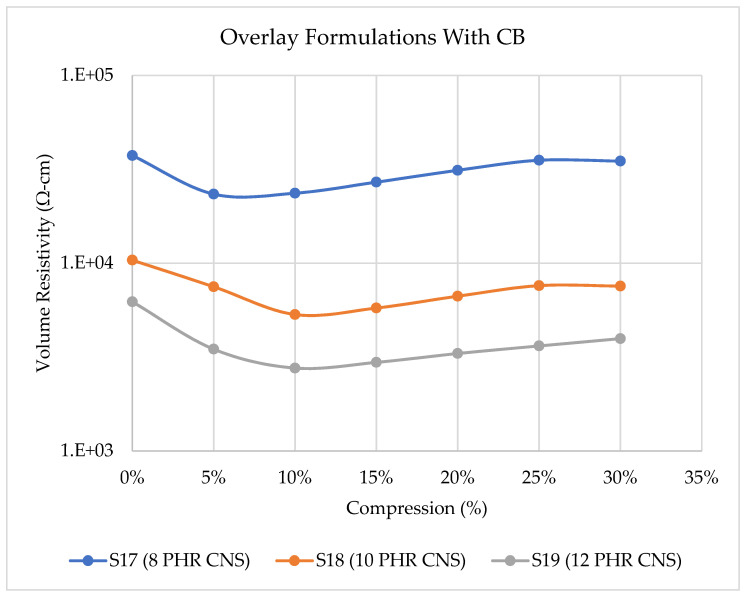
Effect of CNS content within FFKM elastomer O-ring formulations on volume resistivity across multiple compression levels.

**Figure 6 sensors-23-01342-f006:**
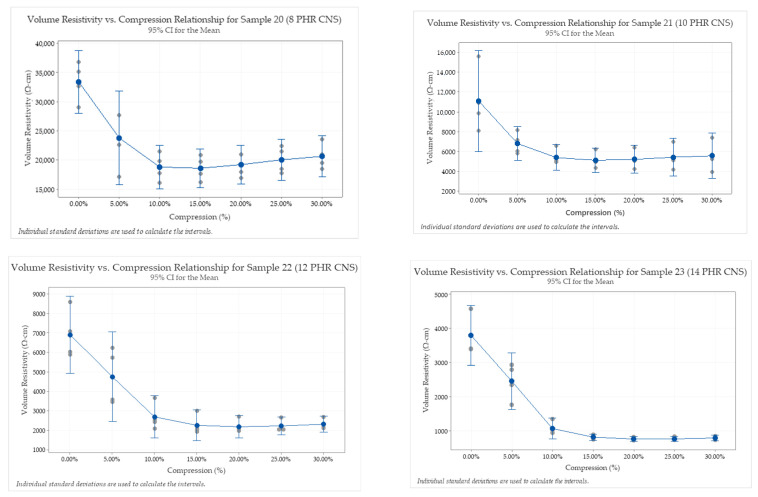
Effect of CNS-filler only, FFKM elastomer O-ring formulations on volume resistivity across multiple compression levels.

**Table 1 sensors-23-01342-t001:** Compositions of the FKM samples 1–9.

Components (PHR: parts per hundred of polymer)	1	2	3	4	5	6	7	8	9
Tecnoflon VPL X 75545 (FKM)	100	100	100	100	100	100	100	100	100
N-990 Black	5	5	5	5	5	5	5	5	5
Austin Black 325	3	3	3	3	3	3	3	3	3
Cabot carbon nanostructure (CNS)	4.8	6.0	7.2	8.4	9.6	4.8	4.0	4.2	4.5
TAIC DLC	5	5	5	5	5	5	5	5	5
Varox DBPH-50	2	2	2	2	2	2	2	2	2
Tensile strength, psi	2149	2602	2859	2925	2962	1900	1847	1863	1935
Elongation, %	136	110	80	77	63	148	158	143	154
Modulus @100%, psi	1847	2538	NA	NA	NA	1556	1460	1522	1535
Modulus @50%, psi	1104	1662	2168	2378	2608	889	843	882	844
Specific Gravity	1.8	1.8	1.8	1.8	1.8	1.8	1.8	1.8	1.8
Hardness, type A, pts	75.8	79.4	83.2	85.2	85.4	76.3	76.0	74.6	75.9
Hardness, type M, pts	79.1	84.0	86.6	88.1	90.5	81.2	78.8	81.8	81.5
Compression Set %	12	13	14	15	16	17	18	19	20
70 h @ 392 °F/200 °C 25% Deflection (avg.)	36.8	41.2	45.6	48.5	51.4	40.0	38.6	38.6	44.3

## Data Availability

Not applicable.

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
