# Peer review of "Smart Quantum Tunneling Composite Sensors to Monitor FKM and FFKM Seals"

_sensors, 2023, doi:10.3390/s23031342_

Round 1

Reviewer 1 Report

see attached pdf.

Author Response

Uploaded file has the answers for Reviewer 1

Reviewer 2 Report

Dear Author,
:
You investigate quantum tunneling elastomeric composites including those formed of curable fluorinated and per-fluorinated compositions, and finished products that incorporate varying levels of carbon nanostructures using these materials for evaluating the performance of the articles in real time when employed in. You monitor monitoring the conductivity of the materials as a function of their level of compression.

The methods are described in a good way. The effects are discussed and the explanations for the results are founded. The important figures are inside. The conclusion is clear. The reference are up todate. Overall it is a good paper.

I miss only the catpions on the figures, please add it. 

BR

The reviewer

Author Response

Uploaded file has the answers to reviewer 2

Reviewer 3 Report

In this work, FKM- and FFKM-based composites were prepared by formulating with carbon nanostructure and other additives, which are used to monitor and predict their life as well as tracking during their use for various applications. The as-prepared composite sensor has great potential application in the future smart elastomeric devices. The paper is interesting and is also well organized. I would like to recommend the acceptance of this manuscript for publication in the journal Sensors after some minor revisions.

(1)     How does the quantum tunneling concept (QTC) affect the preparation and properties of the resulted elastomeric sensors?

(2)     As demonstrated by previous studies (Soft Matter, 2016, 12, 845-852; Mater. Horiz., 2019, 6, 1892-1898.), the geometric distribution of carbon nanofillers in elastomer matrix may significantly influence the electric conductivity and sensitivity of the composite sensors. Therefore, the distribution morphology of carbon nanofillers in the elastomer matrix and its influence on the conductivity and sensitivity need to be investigated.

(3)  One of the most attractive advantages of fluorinated elastomers is the corrosion resistance. How about the corrosion resistance of the as-developed FKM-based sensors? 

Author Response

Uploaded file has the answers for Reviewer 3
